# Host-Directed Therapies Based on Protease Inhibitors to Control *Mycobacterium tuberculosis* and HIV Coinfection

**DOI:** 10.3390/microorganisms13051040

**Published:** 2025-04-30

**Authors:** Manoj Mandal, David Pires, José Miguel Azevedo-Pereira, Elsa Anes

**Affiliations:** 1Host-Pathogen Interactions Unit, Research Institute for Medicines, iMed.ULisboa, Faculty of Pharmacy, Universidade de Lisboa, Av. Prof. Gama Pinto, 1649-003 Lisboa, Portugal; mmandal@ff.ulisboa.pt (M.M.); dpires@ff.ulisboa.pt (D.P.); miguel.pereira@ff.ulisboa.pt (J.M.A.-P.); 2Center for Interdisciplinary Research in Health, Católica Medical School, Universidade Católica Portuguesa, Estrada Octávio Pato, 2635-631 Rio de Mouro, Portugal

**Keywords:** protease inhibitors, saquinavir, cystatins, tuberculosis, HIV coinfection, host-directed therapies

## Abstract

Despite continuous and extensive global efforts in the fight against tuberculosis (TB), this infectious disease continues to exert a tremendous burden on public health concerns and deaths worldwide. TB, caused by the bacterial species *Mycobacterium tuberculosis*, is highly frequent in people living with HIV. The continuing epidemics of both chronic infections and the emergence of antimicrobial resistance, as well as the lack of effective diagnostic tools and drug–drug interactions, pose major challenges in the fight against these pathogens. Developing a wide range of host-directed therapies may improve treatment outcomes, helping alleviate the morbidity and mortality associated with both infections. In this review, we discuss the identification and development of new host-directed strategies based on protease inhibitors and their clinical relevance as adjunctive treatment. In the context of therapeutic agents with novel mechanisms, selective protease inhibitors, including saquinavir (SQV) and cystatins (CstC and CstF), are valuable targets that may provide effective therapeutic solutions for controlling Mtb and HIV coinfection.

## 1. Introduction

Tuberculosis (TB), primarily caused by the bacterial species *Mycobacterium tuberculosis* (Mtb), remains a widespread fatal illness and a major public health problem [1]. In 2023, TB has re-emerged as the world’s leading cause of death from a single infectious agent, displacing coronavirus disease (COVID-19) and surpassing the deaths caused by acquired immunodeficiency syndrome (AIDS) due to human immunodeficiency virus (HIV) [1]. During that year, the total number of deaths due to TB was 1.25 million, accounting for 161,000 deaths among people living with HIV [1]. According to recent estimates, approximately one-quarter of the world’s population has been infected with Mtb [2]. However, not all infected individuals will develop TB. With an appropriate immune response, some people may clear the pathogen or contain the bacilli in a latent TB infection (LTBI). Coinfection with HIV is one of the causes that accelerates progression from LTBI to active disease [3].

Since the advent of combined antiretroviral therapy (cART), HIV infection has become a chronic condition. Continuous cART treatment allows for the suppression and control of the viral load, thereby preventing the transmission of the virus. According to UNAIDS [4], cART coverage reached approximately 76% of diagnosed individuals, with 40 million people living with HIV globally. In 2023, approximately 630,000 people around the world lost their lives to AIDS. It is estimated that at least 1.8 million people have AIDS, a condition usually associated with a late diagnosis and treatment of HIV infection. Recently, AIDS has become more prevalent among individuals who have discontinued cART [5]. In addition to the health risks associated with this, the situation has been shown to increase the risk of HIV transmission and to add to the burden on health systems [6]. TB remains the leading cause of death among people with HIV, responsible for approximately 30% of AIDS-related fatalities.

According to the 2024 WHO Global Tuberculosis Report [1], Mtb is still responsible for an estimated 10.8 million new infections. Furthermore, Mtb has been linked with a significant degree of antimicrobial resistance. The proportion of individuals diagnosed with TB who have rifampicin-resistant TB (RR-TB) and multidrug-resistant TB (MDR-TB, defined as resistance to both rifampicin and isoniazid), collectively referred to as MDR/RR-TB, accounts for approximately 5–10% of the total TB caseload (400,000 cases globally) [1].

A major challenge in the quest for an HIV cure is the emergence of drug resistance during cART, which is predominantly attributed to the high mutation rate of HIV, the prolonged duration of treatment, and inadequate adherence to therapy [7]. The emergence of drug-resistant HIV variants has the consequence of compromising the effective inhibition of viral replication by antiretroviral drugs.

In the present context, the investigation on host-directed therapies that enhance the host responses to control the infection and ameliorate immunopathological damage constitutes a promising strategy to improve disease outcome. This includes the search and repurposing of existing drugs already approved for other conditions that will improve the effectiveness of existing antimicrobials, minimize drug resistance, decrease treatment duration, and adverse effects. In this review, we discuss the identification and development of novel host-directed strategies based on protease inhibitors and their clinical relevance as adjunctive treatment for managing both chronic infections, particularly in the context of a coinfection.

## 2. Immuno-Pathogenesis of Mtb-HIV Coinfection

Mtb is a facultative intracellular pathogen that can reside and replicate within macrophages [8]. These cells are considered host professional phagocytes designed to destroy microorganisms within lysosomes. However, Mtb can subvert this pathway and survive in immature endosomal vesicles [9]. In the initial phases of infection, the surface signatures of Mtb, known as pathogen-associated molecular patterns (PAMPs), are recognized by members of innate pattern recognition receptors (PRRs), expressed by immune cells such as alveolar macrophages and dendritic cells [10]. Upon interaction with PAMPs, there is activation of intracellular signaling pathways, which induce the production of pro-inflammatory cytokines, such as tumor necrosis factor α (TNF-α), interleukin-1 (IL-1), IL-6, IL-12, and chemokines, all relevant in orchestrating the appropriate defense mechanisms [10,11].

The process of Mtb transmission is initiated by the inhalation of respiratory droplets, which subsequently trigger the activation of alveolar macrophages (AM). These phagocytic cells function as the primary line of defense, engulfing the bacteria and thereby containing their dissemination from the lungs to other organs [12]. Pro-inflammatory cytokines released by these immune cells and newly arrived macrophages then initiate the formation of innate granulomas, which represent the earliest mechanism of defense against Mtb [13,14] (Figure 1).

Following the innate phase, the adaptive immune responses are founded in CD4^+^ and CD8^+^ T-lymphocytes that react against Mtb-infected cells within the granuloma structure [15]. These lymphocytes are important for the activation of macrophages, to a more bactericidal M1 inflammatory state, and/or by exerting cytotoxic effects on Mtb-infected cells [10] (Figure 1). Altogether, this leads to the formation of an adaptive granuloma, a structure that is much more effective in containing bacteria in the lungs compared to the initial innate granuloma [15,16]. A caseating granuloma, which is defined by the presence of epithelioid macrophages surrounding a necrotic core consisting of foamy infected macrophages, is a hallmark of TB [15,17]. Besides macrophages, a plethora of cell types have been identified within TB granulomas. These cells include neutrophils, dendritic cells (DCs), B- and T-cells, fibroblasts, natural killer (NK) cells, and cells that secrete components of the extracellular matrix [18]. Uninterrupted secretion of chemokines by activated infected macrophages attracts neutrophils, monocytes, and lymphocytes, thereby continuously feeding the granuloma with new immune cells.

HIV is a master at suppressing the immune response of the host, with the potential to be the major cause of Mtb proliferation and reactivation of LTBI into active TB [19]. Individuals infected with HIV are more susceptible to infection with Mtb, partially attributable to the depletion of CD4^+^ T-cells by apoptosis induced by the viral infection, which consequently alters their effective immune response [20]. In the case of immunocompetent individuals, a concerted response from several immune cells is observed, working together to target Mtb-infected cells. As previously mentioned, CD4^+^ T lymphocytes play a significant role in the control of the infection. CD4^+^ T-cells function as both helper cells, producing interferon γ (IFN-γ) or IL-17 required to control Mtb infection, and as drivers of the formation of cytotoxic cells (CTLs) from CD8^+^ lymphocytes, as well as in B-cells’ cooperation in the production of antibodies [20]. Consequently, the depletion of CD4^+^ T-cells influences the number of activated macrophages, thereby contributing to the persistence and multiplication of Mtb. The overall reduction in CD4^+^ T-lymphocytes, together with the entrance of HIV-infected CD4^+^ T-lymphocytes in granulomas, constitutes a primary factor contributing to the disorganization of the granuloma [21]. This impairment of the granuloma as a solid structure facilitates the easy dissemination of Mtb from the lungs to other organs, thus weakening the immune control over LTBI and enabling the progression to active infection (Figure 1).

Therefore, the coinfection of Mtb with HIV appears to be one of the most significant risk factors for the progression to active TB. As previously mentioned, there is an antibiotic therapy for treating TB and an established antiretroviral therapy (ART) to control chronic HIV infection. Still, the growing resistance to both treatments and the interactions between the various drug classes raise serious concerns about the effective control of these infections [15,20,22,23,24,25,26,27]. Accordingly, there is an imperative requirement for a more profound and detailed understanding of the pathways underlying Mtb-HIV interactions, with an urgent need to develop comprehensive strategies and new efficacious therapeutics to address the challenges posed by TB and TB-HIV coinfection.

## 3. Current Therapeutic Approaches in TB and During HIV Coinfection

The first-line therapeutics for TB are rifampin, isoniazid, pyrazinamide, and ethambutol (RIPE). RIPE therapy is administered over a period of six to nine months, starting with an intensive phase of 2 months, followed by a continuation phase of four or seven months [28]. This well-planned approach effectively addresses TB, reducing the probability of relapses and the emergence of drug-resistant strains. However, in cases involving drug-resistant strains of Mtb, the employment of second-line agents, including injectable aminoglycosides, injectable polypeptides, oral and injectable fluoroquinolones, and bedaquiline, is imperative [29].

Rifampin can be administered either orally or intravenously. It can inhibit Mtb-encoded DNA-dependent RNA polymerase via interaction with the rpoB-encoded β subunit. This interaction prevents RNA synthesis by blocking the elongation of the RNA transcript [30,31]. Consequently, bacterial strains that possess chromosomal mutations in the rpoB gene exhibit resistance to rifampin treatment [31].

Isoniazid is a prodrug activated within the bacteria by the enzyme KatG. This results in the generation of radicals that disrupt the production of mycolic acid, an essential component of the Mtb cell wall [32]. In addition, isoniazid is activated by NADH and InhA, yielding isoniazid-NADH adducts that inhibit InhA, a pivotal enzyme involved in mycolic acid synthesis [33]. Mutations in KatG result in the emergence of resistance to isoniazid [32]. A similar mechanism is observed in the case of pyrazinamide, a prodrug that is activated to pyrazinoic acid by the action of the pyrazinamidase enzyme, encoded by the pncA gene in Mtb [34]. Strains of Mtb that lack pyrazinamidase activity are therefore resistant to pyrazinamide. In contrast, ethambutol exerts its effect by inhibiting arabinosyl transferase, an enzyme that is encoded by the bacterial gene embCAB [35]. Mutations in the embB gene have been associated with resistance to ethambutol, and it has also been demonstrated that ethambutol, when used in conjunction with beta-lactams, enhances their efficacy against Mtb by improving beta-lactams’ access through the cell wall [36].

Strains of Mtb that are resistant to both isoniazid and rifampin (multidrug-resistant tuberculosis—MDR-TB) require treatment with second-line drugs [37]. Extensively drug-resistant tuberculosis (XDR-TB) strains are characterized by resistance to isoniazid, rifampin, a fluoroquinolone, and at least one of the following second-line agents: amikacin, capreomycin, or kanamycin, or otherwise bedaquiline or linezolid [29].

In the context of immunocompromised individuals, such as those infected with HIV, the emergence of drug-resistant strains of Mtb has become a particularly pertinent concern. The management of TB becomes significantly more complex in these cases due to the heightened risk of developing TB in HIV-positive individuals. This imposes regular testing for both LTBI and active TB infection. Another potential complication that should be considered is the occurrence of immune reconstitution inflammatory syndrome (IRIS) in individuals undergoing highly active antiretroviral therapy. This condition is characterized by an uncontrolled hyper-inflammatory response against opportunistic infections, which usually occurs within the first six months of treatment in people living with HIV [38,39,40]. The management of Mtb/HIV coinfection can be attained through a multifaceted approach, including the combination of ARTs and antibiotics, as well as the prevention of IRIS [41]. However, it is crucial to acknowledge the sophistication of this treatment protocol and the potential for drug–drug interactions (DDIs). Furthermore, the duration of treatment should be taken into consideration, with shorter regimens being preferred. In addition, TB prophylactic treatment for individuals with HIV should be customized in accordance with their cART regimen to enhance the effectiveness of the treatment and to limit the adverse effects associated with the coinfection [42].

The Centers for Disease Control (CDC) currently recommends a twelve-week, once-weekly regimen of isoniazid and rifapentine with cART in the absence of any DDIs [42]. Nonetheless, it is imperative to take special consideration when prescribing this combination due to its potential DDIs. This includes drugs like rifapentine and rifampin, which are contraindicated in HIV-positive individuals taking all protease inhibitors (doravirine, rilpivirine, bictegravir, cabotegravir, elvitegravir, temsavir, and lenacapavir) [42,43]. Appropriate cART includes efavirenz once daily or raltegravir twice daily with either abacavir/lamivudine or tenofovir disoproxil fumarate/emtricitabine. However, rifampin and rifapentine significantly reduce the effective dose concentration of these drugs, thus necessitating adjustments to ART regimens, which in turn increase the likelihood of drug-related adverse effects [42].

The adjustment in dosage of drugs is essential for the maintenance of adequate drug levels, which will provide efficient treatment for both TB and HIV without any risk of failure. For instance, the current guidelines recommend a dose of 600 mg of rifamycin to be given alone or co-administered with efavirenz [44], but some sources suggested a higher dose of 800 mg for individuals with a body weight of more than 60 kg [44]. It is important to note that dolutegravir is not recommended for patients with integrase strand transfer inhibitor resistance, though it can also be used as a first-line ART [42]. Rifabutin, another drug belonging to the rifamycin family, has demonstrated its efficacy in the treatment of TB [45]. An alternative option that has gained popularity is the utilization of shorter drug courses, which have been shown to exhibit higher efficiency and greater completion rates compared to traditional regimens [46]. The BRIEF TB/AF279 clinical trial demonstrated the efficacy of an ultra-short regimen comprising a one-month course of rifapentine and isoniazid, achieving a higher completion rate than the nine-month course of isoniazid in preventing TB infection in HIV-infected individuals [47]. This regimen is now recommended by the WHO [47,48]. In addition, the development of several compounds has also been reported, including a family of N-alkyl nitrobenzamides that exhibit promising antitubercular activities and can even be classified as a structural simplification of previously known inhibitors of decaprenylphosphoryl-β-D-ribofuranose 2′-oxidase (DprE1), a critical enzyme of Mtb and a novel antitubercular target [49].

In HIV-positive patients with TB, CD4^+^ T-cell counts are a crucial factor in determining the initiation of cART [42]. Based on the CD4^+^ T-cell counts, cART should be initiated within two weeks of TB treatment initiation if the CD4^+^ T-cell count is less than 50 cells/mm^3^, while it can be commenced within eight weeks of TB treatment initiation if the CD4^+^ T-cell count is 50 cells/mm^3^ or above [42,43]. Two main treatment options are currently available for individuals with TB and HIV infection. The first option involves a six-to-nine-month period of intensive treatment with a combination of drugs, specifically rifampin, isoniazid, pyrazinamide, and ethambutol for two months, followed by a four-month continuation phase of rifampin and isoniazid [50]. The second option consists of a 4-month regimen of rifapentine-moxifloxacin, which can be administered to patients with a CD4^+^ T-cell count of 100 cells/microliter, and an ART regimen including efavirenz [51]. Additionally, a number of innovative therapeutic approaches for countering the coinfection have been proposed, including the use of dual-targeted anti-HIV/anti-TB heterodimers [52] and the targeting of HIV’s inhibition of NK cell-mediated immunity by HIV in response to Mtb infection [53].

Despite the advancements and developments in therapeutic approaches to treat and control TB and chronic HIV infection, the synergistic action of these pathogens, rising resistance to both treatments, and drug–drug interactions are leading to significant diagnostic and therapeutic challenges. Coinfection with Mtb increases the risk of death among HIV-positive patients, while late-stage HIV infection increases the risk of developing TB in latently infected individuals. After decades without signs of progress, the drug development pipeline for tuberculosis has finally seen the introduction of new antibiotics in clinical practice. The BPaL regimen, consisting of bedaquiline, pretomanid, and linezolid, has been recently recommended to treat drug-resistant TB [54]. The regimen lasts between 6 and 24 months, depending on resistance levels and response to treatment. This regimen has produced remarkable results in clinical trials, showing comparable results to the standard treatment when applied to drug-susceptible TB cases and high rates of success in treating XDR TB and unresponsive MDR TB, even in HIV-positive individuals [55]. Still, it suffers from the same drawbacks of the standard regimen, with several treated individuals showing adverse effects to the treatment, including 81% of the patients reporting peripheral neuropathy and approximately half the patients showing evidence of hematologic toxic effects. The side effects associated with the standard RIPE regimen and the different drug-resistant TB regimens have been extensively documented. Severe side effects related to the termination of standard treatment can include hepatotoxicity, peripheral neuropathy, exanthema, gastrointestinal disorders, and arthralgia, among others [56,57,58,59]. In MDR TB treatment, these are further complicated, with a study reporting gastrointestinal disturbance (18.4%), psychiatric disorder (5.5%), arthralgia (4.7%), hepatitis (3.9%), peripheral neuropathy (3.1%), hypothyroidism (2.3%), epileptic seizures (2%), dermatological effects (2%), ototoxicity (1.6%), and nephrotoxicity (1.2%) in the enrolled patients [60]. These adverse effects pose additional ethical questions when considering preventive therapy offered to individuals presumptively infected with Mtb. The preventive regimens have seen substantial length reductions from six to nine months to one to four months [61]. Yet, these are still long treatments in absolute terms, especially considering the low baseline risk of developing tuberculosis. In addition to these side effects, cases of bedaquiline resistance are rising in Africa following first treatment, providing evidence for a careful usage of this last available regimen for XDRTB, for which currently no good alternative exists [62].

Even accounting for the high success rates associated with the situation of full compliance to therapy, the concomitant costs, aggravated in cases of MDR and XDR TB, constitute a major challenge in high-TB-burden countries. According to the WHO, the percentage of TB patients and their households that experienced catastrophic total costs associated with TB and its treatment (defined as >20% of annual household expenditure or income) ranged from 13% to 92%, averaging at 49% [63]. All these factors highlight an ongoing and critical demand for complementary management approaches that can bypass the constraints of the existing antimicrobial treatment portfolio, addressing the problems of drug resistance, drug interaction, adverse drug effects, access to and cost of treatment, and patient compliance. Host-directed therapies, particularly those developed through drug repurposing, have the potential to fill this role as a complementary strategy to improve TB treatment.

## 4. Host-Directed Therapies: Current Status and Recent Progress for the Treatment of Infections

From an evolutionary perspective, host-pathogen interactions can be regarded as states in which microbes exist without causing harm or manifesting overt disease [64,65]. The action of the host’s innate and adaptive immune surveillance mechanisms determines whether the infection will be contained or whether it will progress to clinical disease, resulting in recovery or death [64]. It is imperative to acknowledge the crucial function of host-related events in the efficacy of antimicrobial treatment, particularly because disease progression subsequent to infection can result in tissue damage, which can lead to long-term functional impairment and increased mortality. The underlying mechanisms of this process are multifaceted and include, but are not limited to, immune dysregulation resulting from various etiologies, such as an excessive inflammatory response to infection, the use of immunosuppressive therapies, disadvantageous socioeconomic conditions, HIV infection, or the presence of comorbidity, including non-communicable diseases such as diabetes, neoplasia, and chronic pulmonary disease [66].

In the context of infectious diseases, it is crucial to understand the complex interplay between the host and the pathogen to elucidate the mechanisms underlying pathogenesis. In the past few years, there has been a paradigm shift in scientific research, with a transition from a direct targeting of the pathogen to a focus on host factors. This paradigm shift has opened promising new avenues that hold the potential to enhance clinical outcomes, referred to as host-directed therapies. Host-directed therapies (HDTs) are defined as host-directed interventions that can modulate intracellular pathways involved in innate or adaptive immune responses to microbes to enhance immune response and reduce immunopathology [67]. About 90% of the tested HDTs have demonstrated comparable efficacy levels towards drug-resistant and drug-susceptible pathogens [68]. In addition to their efficacy in battling drug-resistant pathogens, HDTs have been hypothesized to reduce the likelihood of developing drug resistance by targeting multiple cellular and intracellular mechanisms that are critical for microbial replication and pathogenesis [68].

HDTs can also enhance the efficacy of anti-microbial treatments. A recent study evaluated 183 host-directed drugs, of which 30% were found to have activity against more than one pathogen. The majority of these (52%) targeted close evolutionary relatives, while 48% were active against evolutionarily distinct groups at the levels of families, kingdoms, and domains [68,69]. The HDTs comprise drugs with different mechanisms of action. Among these agents are those that enhance host immunity, such as CAR-T cells, which have been used in the therapeutic management of HIV-1-infected individuals, leading to a transient reduction in the HIV-1 viral reservoir [70]. The administration of type I or III interferon (IFN) has also been used for the treatment of both SARS-CoV-2 infection and chronic hepatitis C (HCV) infection, respectively [71,72,73]; N-acetylcysteine, an antioxidant, has been utilized to protect cells from oxidative damage in TB [74]; anti-pathogen antibodies have been used for the treatment of SARS-CoV-2 [75] and respiratory syncytial virus (RSV) infections [76].

The immunopathology associated with the immune response to the pathogen can be mitigated by HDTs that attenuate inflammation. These HDTs comprise compounds that target cytokines, such as the interleukin (IL)-6 receptor blockade in COVID-19 [77], tumor necrosis factor (TNF) treatment in cases of TB meningitis [78], and the Janus kinase (JAK) inhibitors in COVID-19 [77]. A number of other potential anti-inflammatory HDTs have been identified, including the use of corticosteroids [77,79], antioxidants such as N-acetylcysteine [74], vitamin D, statins, and cyclooxygenase 2 inhibitors [80,81,82].

HDTs are also expected to modulate the immune response to Mtb, thus emerging as a compelling antibiotic adjuvant therapeutic strategy. There is mounting evidence to suggest that several facets of the immune response, including reactive oxygen species production, antimicrobial peptide synthesis, cytokine production, autophagy induction, and cell-mediated immunity, can be modulated by HDT to hinder Mtb growth (Figure 2A). One potential host therapeutic target against Mtb is the impairment of granuloma structure through TNF-α blockade by using etanercept (Enbrel, a soluble TNF-α receptor). This disruption enhances antibiotic access to the bacteria and curtails lung pathology [83] (Figure 1). Another recent clinical trial has demonstrated that etanercept-mediated TNF blockade increases CD4^+^ cells amongst patients with HIV-associated TB in the early treatment phase of TB. Nevertheless, the administration of anti-TNF-α therapy has been demonstrated to exacerbate the severity of the disease by suppressing the host’s immune response, a finding that has been established in animal models [83,84].

Furthermore, HDTs can modulate immune responses by targeting the vitamin D signaling pathway, which is critical for enhancing the Mtb-killing activity of macrophages. Vitamin D has been shown to induce autophagy by downregulating the mTOR protein [85]. Additionally, CAMP gene expression can be activated by the co-administration of phenylbutyrate with vitamin D_3_ and standard anti-TB drug therapy and has been shown to inhibit the intracellular growth of Mtb [86,87]. A recent study revealed that the supplementation of vitamin D at a dose of 2.5 mg in combination with standard first-line anti-TB drugs effectively rectified the vitamin D deficiency. Still, no significant change in the Mtb culture conversion in pulmonary TB has been observed [88,89]. A similar finding was reported for sputum conversion, which was enhanced with vitamin D supplementation in TB patients with the vitamin D receptor “tt” genotype [89]. These conflicting results may be attributable to variations in dose, treatment duration, disease stage, and other factors. Therefore, further studies with an adequate sample size and appropriate clinical conditions are required to understand the epidemiological effects of vitamin D supplementation in TB.

The potential of HDTs as an adjuvant in the treatment of TB also includes drugs that target the anti-inflammatory response, such as aspirin and ibuprofen [90] (Figure 2A). A study conducted in the C3H4B/Fej murine model demonstrated that aspirin exhibits a greater effect on bacillary load, resulting in enhanced survival in advanced disease [91]. In a different study, the administration of aspirin to BALB/c mice led to a potentiation of the action of pyrazinamide, resulting in enhanced clearance of viable mycobacteria in both the spleen and lungs. However, the combination of aspirin and isoniazid resulted in increased bacterial loads [92]. These findings collectively indicate that further clinical studies should be conducted to evaluate the therapeutic effect of concomitant aspirin administration when combined with other anti-TB drug combinations. In a mouse model mimicking active TB, the bacillary load was also reduced by ibuprofen treatment [93]. Additionally, in a different study conducted using a murine model, ibuprofen was found to enhance the efficacy of pyrazinamide in TB treatment [92].

It is anticipated that HDTs should improve cell-mediated immunity. Such agents include cholesterol-reducing drugs such as statins. Several mechanisms against TB have been identified, which include the inhibition of foamy cell formation, the support of Mtb persistence via the reversal of cholesterol biosynthesis [94], and the induction of phagosomal maturation and autophagy [95]. For instance, the autophagy inhibitor 3-methyladenine has been shown to reverse autophagy and the maturation of phagosomes. In contrast, the administration of the statin-simvastatin at a dose of 25 mg/kg has been shown to accelerate the clearance of bacteria in the lungs of mice when used in conjunction with standard therapy [96,97]. However, an extensive retrospective study utilizing a national medical claims database revealed that statins did not offer protection against tuberculosis in patients with diabetes mellitus [97,98], highlighting the need for further research to determine the most effective agent and dosing schedule for clinical trials. Other mechanisms involve an increase in the percentage of natural killer T (NKT) cells in the cultures and upregulation of co-stimulatory molecules on monocytes, along with higher IL-1β and IL-12p70 secretion [99,100] and inhibition of TGF-β [101,102].

Various evidence and studies, as well as the number of ongoing interventional clinical trials, reveal HDTs as a promising solution by repurposing known effective drugs and targeting new candidates. The results from these trials will be relevant to determine their potential beyond pre-clinical evidence. Repurposed drugs for HDTs have the benefit of decades of clinical application. Still, evidence must be collected concerning their safety and bioavailability in TB patients that might show heightened susceptibility to adverse effects, while also experiencing interactions with the antimicrobial therapy. As mentioned before, hepatic and kidney disorders are already described as significant adverse drug effects and reasons for patient non-compliance. The addition of further drugs might introduce an additional burden to the patient’s liver and kidneys. One must also consider the diverse population that is affected by TB. Many repurposed drugs and candidate HDTs are studied in cohorts from high-income countries that may not reflect the populations affected by TB and HIV from high-burden countries. This is particularly relevant for severe manifestations of TB, such as meningitis and miliary TB, which are more often found in those regions. The following section will be based on new strategies using novel host-directed strategies identified and developed through protease inhibition (Figure 2B).

## 5. Protease Inhibitors as a Strategy to Control Infectious Diseases

Enzymes have long been recognized as ideal candidates for target-based drug development, as evidenced by decades of research on disease agent proteases. Screening for selective protease inhibitors has been a valuable strategy to deliver effective therapeutic solutions for treating some of the most important infectious diseases like malaria and COVID-19 [103]. A number of proteolytic targets for infectious diseases have been validated, including proteasomes of the malarial pathogen *Plasmodium falciparum*, *P. falciparum* aspartyl proteases, and SARS-CoV-2 viral proteases [103]. Furthermore, low-molecular-weight inhibitors of proteases have recently emerged as promising therapeutic agents for treating numerous diseases, including hypertension, diabetes, and various cancers [104].

Indeed, proteases play crucial roles in nearly all biological processes, both within individual organisms and in extracellular events. This generally involves either the activation of other enzymes through their well-timed processing, resulting in the generation of active, accessible catalytic sites on substrates, or conversely, the inactivation of proteins through their proteolytic fragmentation [105]. Proteases are classified according to the composition of their active site and catalytic reaction. To date, seven classes of proteases have been identified, which include metallo-, serine, aspartyl, cysteine, threonine, glutamate, and asparagine proteases [106]. Among the five most abundant, serine proteolytic enzymes are found in the highest natural abundance, followed in decreasing order by metallo-, cysteine-, aspartate, and threonine proteases [107]. These are the five classes that have been identified in humans, accounting for approximately 3% of the genome [108] and participating in the regulation of a variety of biological processes, including ovulation, embryogenesis, organ formation, tissue remodeling, immune response, antigen presentation, pathogen infection, cell death, cell migration/invasion, cell signaling, and wound healing [109,110,111].

Proteolytic enzymes are regarded as a primary component of the major virulent factors of infectious agents, playing a pivotal role in their development, reproduction, and interactions with host/invertebrate vector tissues. Therefore, these enzymes are considered promising targets for designing new drug candidates for the treatment of infectious diseases [112,113]. In 1995, the discovery of HIV protease inhibitors (HIV-PIs), such as saquinavir, lopinavir, and ritonavir [114,115], demonstrated the antiviral efficacy of enzyme-targeted drugs. These HIV-PIs function by selectively blocking the retroviral protease, thereby preventing the processing of the long polypeptide that the viral RNA genome encodes into its individual viral proteins [116]. The advent of novel inhibitors and their combinations has been fundamental to the development of effective and less toxic antiretroviral therapy, thereby transforming HIV infection from a fatal to a more manageable disease. The development of protease inhibition-based treatment has emerged as an attractive and potentially efficacious strategy against infections since the discovery and approval of HIV-PIs. For instance, the papain-like cysteine protease cruzain has been demonstrated to be essential for the life cycle and virulence of *Trypanosoma cruzi*, the causative agent of Chagas disease [117]. Likewise, an irreversible cruzain inhibitor, vinyl-sulfone, has demonstrated efficacy against schistosomiasis, hookworm infections, and cryptosporidiosis [117]. In the context of treating multiple myeloma, selective proteasome inhibitors such as carfilzomib, bortezomib, and ixazomib have been developed as a therapeutic strategy [118]. In addition, the development of specific inhibitors is critical in the treatment of infectious diseases such as malaria, leishmaniasis, schistosomiasis, and Chagas disease [119].

Another example that substantiates the efficacy of protease-inhibition-based drug development is evident in the repurposing of drugs designed to target SARS-CoV-2. These candidate compounds have been shown to inhibit the SARS-CoV-2 main protease (M^pro^), which is known to play a crucial role in the viral replication process within infected cells [120]. The PL^pro^ protease of SARS-CoV-2 shares a high degree of sequence and structural similarity with PL^pro^ of the previously emerged coronavirus (SARS-CoV-1). It has been characterized as displaying analogous functions in virus replication and modulation of the host’s immune responses [121]. Thus, the inhibition of SARS-CoV-2 protease can block viral replication and enhance the innate immune responses in acute cases of SARS-CoV-2 infection. Several studies have demonstrated the efficacy of repurposing drugs developed for the treatment of SARS-CoV-1-PL^pro^ to inhibit SARS-CoV-2-PL^pro^ [121,122,123].

Regarding tuberculosis, the Mtb proteasome has been the subject of recent studies probing protease inhibitors in compound libraries for potential interactions with mycobacterial proteases [124,125,126]. Mtb caseinolytic proteases (Clp) have been explored as such potential targets. These serine proteases play important roles in proteostasis of eukaryotic and prokaryotic cells and are often associated with infectivity and virulence of several pathogens. The Clp system comprises the highly conserved caseinolytic protease P (ClpP) and AAA + (ATPases associated with diverse cellular activities) chaperone-unfoldases [127]. Mtb uniquely harbors two ClpP paralogs—ClpP1 and ClpP2—that function together as a heterotetradecameric complex with an essential role in maintaining protein homeostasis, degrading misfolded or damaged proteins, and modulating virulence factors [128]. These ClpP are essential for Mtb survival in vitro and during infection [129], making them attractive and validated drug targets [130,131]. Several small-molecule inhibitors, such as boronic acid derivatives and cediranib-based compounds, have shown potent activity against MtbClpP1P2. Cediranib, originally developed as a human VEGFR-2 inhibitor for cancer treatment, was recently identified as a potent allosteric inhibitor of MtbClpP1P2, disrupting its activity and effectively suppressing Mtb growth [130]. Bortezomib, a clinically used human proteasome inhibitor for multiple myeloma, has been identified as a potent inhibitor of mycobacterial ClpP1P2 using a target mechanism-based whole-cell screening approach [132]. This compound showed growth inhibition and microbicidal activity in Mtb. However, bortezomib’s lack of selectivity toward human proteasomes presents a major obstacle for its use as an anti-TB drug. Many ClpP inhibitors face challenges due to cross-reactivity with human mitochondrial ClpP, which can lead to mitochondrial dysfunction. Finding compounds designed to retain anti-Mtb activity while significantly reducing off-target effects on the human proteasome is paramount to improving their safety profile [131]. To address this, analogs with modified warheads, such as chloromethyl ketones, have been developed to retain anti-mycobacterial ClpP1P2 activity while significantly reducing off-target effects on human cells [133]. These analogs demonstrate dual inhibition of both the ClpP1P2 and the Mtb proteasome, a combination that appears particularly lethal to Mtb yet spares human proteasome function. Additionally, efforts to inhibit related components such as ClpC1, the AAA+ chaperone-unfoldase that cooperates with ClpP1P2, have also yielded promising quinoline-based inhibitors with potent antimycobacterial activity [131].

As previously mentioned, saquinavir (SQV) controls HIV infection by interfering with HIV protease activity, ultimately preventing virion infectivity and intercellular transfer. It was hypothesized that SQV could be repurposed to regulate a variety of protease-dependent infectious organisms. The proteostasis network, comprising chaperones, proteases, and the proteasome, plays a major role in the survival of Mtb under cognitive host immune stress [134]. It has been demonstrated that HIV and Mtb interact synergistically, modifying the host immune environment, thereby promoting the replication and dissemination of both pathogens. Hence, it is plausible that during coinfection, HIV PI could interfere with the host proteases. The potential and proven proteolytic targets against various infectious diseases provide compelling evidence for establishing protease inhibitors as a drug development platform and other therapeutic strategies. A more profound understanding of microbial dependence on cellular proteases and their inhibitors may establish a strong basis for designing novel host-directed strategies to control microbial infection.

## 6. New Host-Directed Strategies Based on Protease Inhibitors for Mtb and Coinfection with HIV

This section will focus on molecules and strategies targeting host proteases, which have been putatively associated with proteolytic activity in Mtb-infected macrophages and coinfection with HIV. Previous studies have already demonstrated that Mtb induces a general down-regulation of lysosomal proteases (cathepsins B, S, and L), which ultimately helps Mtb to establish intracellular niches within macrophages [135,136]. In the context of coinfection, the synergistic effect between HIV and Mtb leads to the emergence of these niches as important mediators of the infection’s pathogenicity. In this scenario, it is possible that HIV PIs, including SQV, could be repurposed to control infections caused by these protease-dependent pathogens. This hypothesis is further substantiated by the fact that SQV and other PIs have already shown inhibitory effects against various pathogens, including *Plasmodium falciparum* [137], *Trypanosoma cruzei* [138], and SARS-CoV [139].

### 6.1. Repurposing Saquinavir as a Host-Directed Strategy to Control Mycobacterium tuberculosis Infection

A plethora of antiretroviral drugs are already available for the treatment of HIV infection. However, only a few of these (PIs) can interfere with virus production and release from macrophages during chronic infection. It has been demonstrated that HIV PIs, particularly SQV, may interfere with the proteases of the host during Mtb or Mtb/HIV infection. In this particular context, SQV, a first-generation HIV PI, has been repurposed as a host-directed therapy for tuberculosis [140]. SQV, rather than acting directly on infected macrophages as a PI, enhances the proteolytic activity of omnicathepsins during Mtb infection or during coinfection. Among endolysosomal enzymes, cathepsin S was the most affected by SQV with an induced proteolytic activity, at a broad pH range [140]. Cathepsin S is also involved in the processing of antigen presentation and the regulation of autophagy [140,141,142,143]. Consequently, the study [140] demonstrated that SQV treatment led to enhanced pathogen killing attributable to heightened cathepsin activity. The increased activity of cathepsin S in Mtb-infected cells treated with SQV led to a marked upregulation of HLA class II molecules and resulted in boosted levels of T-cell-secreted IFN-γ [116].

Cathepsins S and L are indeed involved in the process of autophagy, an innate response impaired by Mtb and HIV [144]. Interestingly, upon infection by one pathogen, alveolar macrophages may preferentially sustain the second pathogen or may even induce a similar response in neighboring cells [145]. Therefore, the SQV-mediated enhancement of proteolytic activities of cathepsins S and L, as confirmed in the study by Pires et al. [140], would also augment autophagy. Consequently, this would facilitate infected cells in the clearance of pathogens, as well as inflammatory signaling mediators, ultimately resulting in the reduction in tissue inflammation [146]. Furthermore, the study also demonstrated enhanced internalization of SQV in macrophages using a liposomal drug delivery system. This system significantly impacts the proteolytic activity of cathepsins compared to that of free drug treatment, and without any cytotoxicity [147]. Advanced drug-delivery methods are of paramount importance for SQV, as its serum availability is low due to first-pass metabolism by the liver [148]. This is further worsened by the standard anti-TB drug rifampicin, which activates liver metabolism of SQV [149]. In past TB-HIV regimens, another protease inhibitor, ritonavir, would be introduced to counter the adverse effects of rifampicin and improve SQV serum levels. In conclusion, studies and relevant data support SQV’s potential as a host-directed therapy for TB. Still, the use of SQV would require demonstration that its use in conjunction with targeted drug-delivery methods can overcome negative interactions with rifampicin and increase its bioavailability. This would significantly increase its potential, since SQV’s role as an enhancer of macrophages’ killing activity would be best applied in conjunction with the antimicrobial therapy.

### 6.2. Modulation of Cystatins C and F as a Host-Directed Strategy to Control Mtb Mono-Infection or Coinfection with HIV

Several proteolytic enzymes are involved in various physiological processes that are related to the maintenance of homeostasis in host cells [150]. These enzymes have been shown to play an important role in the proteolytic killing of bacteria that have been taken up via the process of phagocytosis [151]. Mtb is a facultative intracellular pathogen with the potential to establish its primary niche within early endolysosomal vesicles. To this end, the bacilli hamper phagosomal maturation and subsequent intracellular macrophage efficacy [152,153]. This dysregulation of host macrophages can affect the expression of proteolytic enzymes, such as lysosomal cathepsins, leading to an enhanced intracellular survival of Mtb [135]. The abnormal activity of proteolytic enzymes must be tightly controlled, as it can result in severe dysfunction and pathology. Cystatins (Csts) are a well-known class of endogenous modulators of cathepsins, which generally control their excessive enzymatic activity by binding and sequestering them within cells, tissues, and body fluids [150]. Pathogens, such as Mtb and HIV, could evolve strategies for manipulating these early events to prevent the activation of the microbicidal mechanisms, allowing their survival or replication within the cells. Therefore, Csts may emerge as potential targets that could be manipulated to restore cathepsin activity in the context of Mtb infection or coinfection with HIV.

Csts are well-characterized inhibitors that are further divided into four different families. Type I Csts (stefins) consist of CstA and B and exist within the cytosol and nucleus, while type II Csts include CstEM, S, SA, SN, and D, which are secreted and function as extracellular proteins [154,155]. A subset of the secreted type II Csts (CstC and F) have been observed to be internalized by immune cells or to translocate from the secretory pathway, translocating into the cytosol of the immune cells, accumulating in endosomal/lysosomal vesicles [156,157]. Type III Csts include kininogens that circulate in the blood as precursors of the vasoactive peptide kinin (Cst families reviewed in [154,155]). Finally, type IV Csts comprise thyropines and are characterized as a Csts family originating from the observation that a fragment of the p41 invariant chain bound to MHC class II molecules inhibits cathepsin L [158,159]. Cathepsins and their natural inhibitors are regarded as the key players in Mtb and HIV infection. Thus, therapies based on manipulating these relevant players could restore the protease/antiprotease balance, which in turn can control infection, transmission, and excessive inflammatory responses. A transcriptional mapping of the human macrophage transcriptome for type I and type II cystatins during monoinfection or coinfection revealed CstC as a significant target and CstF as a potential target for controlling Mtb or Mtb/HIV infection [160]. The study demonstrated that knocking down CstC mRNA significantly improved the intracellular killing of Mtb. Furthermore, an increase in the expression of human leukocyte antigen (HLA) class II in macrophages, the proliferation of CD4+ T-lymphocytes, and IFN-γ secretion was observed [160]. This increased IFN-γ production will result in the activation of macrophages with proinflammatory properties and the increase in microbicidal activities against Mtb [151]. A subsequent study has demonstrated the development of a new potential nanomedicine based on chitosan for delivering cystatin C-targeting siRNAs to the infected macrophage model with significant impact on the intracellular viability of Mtb [161]. The results showed that manipulating CstC expression in human macrophages substantially affected the innate immune regulation of Mtb intracellular growth and the strengthening of the adaptive immune responses against mycobacterial infection [160].

Furthermore, the depletion of CstF has been demonstrated to modulate the proteolytic activity of the macrophages’ lysosomal cathepsins (CtsS, CtsL, and CtsB), thereby enhancing the intracellular killing of Mtb [162]. Indeed, the depletion of CstF expression has been shown to control Mtb intracellular loads, including multidrug-resistant clinical strains resistant to first-line antibiotics used to treat TB (Figure 3).

These findings are consistent with earlier observations using the CstC cathepsin inhibitor, where similar results were obtained in macrophages [160]. Direct regulation for cathepsins L and S was indeed demonstrated via targeting cystatin F expression, as previously reported, while no evidence for direct inhibition of cathepsin B was found [160,163]. The findings of this study do not differentiate between a direct or indirect role for CstF, which may be the reason for the contradictory results obtained. Overall, these two studies on Cst C and Cst F indicate that targeting protease inhibitors could be a promising approach to improve the control of Mtb infections, even in clinical strains that are resistant to first-line antibiotics used to treat TB [162,164].

HIV-infected patients frequently exhibit pleural effusion due to TB in the context of coinfection [140]. Furthermore, higher levels of CstF have been detected in pleural effusion of TB patients compared to other inflammatory conditions [165]. Consequently, CstF could be anticipated as a potential target in the context of Mtb/HIV coinfection. It has been hypothesized that the depletion of CstF could augment control over HIV infection, in addition to the already established microbicidal effects over Mtb [166]. Indeed, the analysis revealed that CstF depletion in Mtb-infected macrophages enhances cathepsin C activity in cocultured lymphocytes infected with HIV, which in turn augments their granzyme cytotoxic effects. Consequently, a significant impact was observed on HIV replication and viral loads [166] (Figure 4). The study proposed that CstF could serve as a novel target for new therapeutic strategies aimed at controlling both Mtb and HIV pathogens.

### 6.3. Matrix Metalloproteinase Inhibitors to Treat TB Immunopathology

Matrix metalloproteinases (MMPs) are zinc-dependent enzymes that degrade extracellular matrix components like collagen and elastin. MMPs play a key role in TB pathology by degrading the pulmonary extracellular matrix and driving tissue destruction [167,168]. In active TB, macrophage MMP-1 and neutrophil MMP-8 are strongly upregulated and contribute to lung cavitation by driving collagen degradation in the alveoli [169,170]. Higher levels of these MMPs correlate with increased lung damage in TB, and their activity is pronounced in lung lesions, particularly MMP-1, which is the most upregulated protease in the sputum and lungs of TB patients [169,171]. MMP-mediated damage is not confined to the lungs. In TB meningitis, MMPs contribute to blood-brain barrier breakdown and brain tissue damage, which is linked to poor neurological outcomes [167]. By contrast, in HIV coinfection, MMP levels are significantly lower in the sputum [172], and coinfected patients develop fewer or smaller cavitary lung lesions [173]. HIV coinfection increases the risk of miliary (disseminated) TB, and, paradoxically, in contrast to what occurs in the lungs, higher systemic levels of collagen degradation products in coinfected patients point towards systemic MMP dysregulation [170,171]. MMP-8 was found particularly elevated in coinfected patients. These alterations become more evident when HIV patients initiate cART and develop TB immune reconstitution inflammatory syndrome (TB-IRIS), an acute inflammatory response to Mtb. MMP levels were shown to increase during TB-IRIS, with neutrophil-derived MMP-8 appearing to mediate lung injury in IRIS patients, showing correlated increases in neutrophil counts and collagen breakdown [171].

MMPs are attractive targets for an HDT that limits MMP-driven TB immunopathology. Multiple candidates have been explored in pre-clinical models. Repurposed broad-spectrum MMP inhibitors, such as the small molecules batimastat, marimastat, and cipemastat, have been tested in animal models of TB with mixed results and toxicity concerns. Batimastat and marimastat treatment alone showed no effects on Mtb burden in a mouse model, yet in combination with isoniazid and rifampicin, MMP inhibition improved drug delivery into TB lesions and boosted the bacteria killing of the standard drugs [174]. Marimastat, in conjunction with isoniazid, further preserved lung tissue integrity. On the contrary, cipemastat treatment in mouse and rabbit models of cavitary pulmonary TB showed worse cavitation and more disease severity [175,176].

In a different study targeting MMP-9, a protease associated with macrophage recruitment and granuloma formation, the researchers used an anti-MMP-9 antibody combined with rifampin, isoniazid, and pyrazinamide, showing reduced bacterial burden in a mouse model of TB compared to the antibiotic treatment alone [177]. However, treatment with a control isotype antibody produced a similar benefit. Another specific MMP-9 inhibitor, SB-3CT, has also shown beneficial activity combined with the same standard antibiotics, but in a mouse model of TB meningitis [178]. SB-3CT demonstrated decreased levels of MMP-9 in the serum and tissues of the infected mice and a reduction in the bacterial burden in some tissues. In the same study, dexamethasone also decreased the elevated levels of MMP-9 in sera and tissues of the infected mice. However, dexamethasone administration had an inhibitory effect on bacillary clearance, while SB-3CT potentiated the bacillary clearance, suggesting that specific inhibition of MMP-9 can be more beneficial in the management of TB meningitis.

Doxycycline, a tetracycline antibiotic, inhibits several MMPs, including MMP-1, -8, -9, and others. A preclinical study provided proof-of-concept that doxycycline can reduce TB pathology [170]. In vitro, doxycycline treatment of Mtb-infected macrophages and epithelial cells significantly suppressed MMP-1, -3, and -9 secretion by inhibiting their gene promoter activation. Doxycycline also showed bacteriostatic properties in broth culture. In a guinea pig TB model, doxycycline led to a dose-dependent reduction in lung mycobacterial burden over 8 weeks. These findings suggest that doxycycline may confer a dual benefit: limiting host tissue destruction and restraining bacterial proliferation. Importantly, doxycycline possesses broad anti-inflammatory effects, which can confound its MMP-specific activity with a wider dampening of TB-related inflammation [179,180]. One advantage of doxycycline is its low cost and wide availability, making it a practical candidate for HDT. A recent phase II clinical trial investigated doxycycline as an adjunctive therapy in pulmonary TB [181]. The trial demonstrated that 14 days of doxycycline (given alongside standard TB antibiotics) can safely dampen MMP activity and inflammation: patients receiving doxycycline showed rapid decreases in sputum MMP-1, -8, -9, -12, and -13 levels, accompanied by reduced collagen and elastin degradation and even a reduction in cavity size. Importantly, this MMP suppression did not impair bacterial clearance, as sputum culture conversion and mycobacterial loads were similar between groups. These findings highlight doxycycline’s potential as a host-directed therapy to limit TB immunopathology, and larger trials are underway to confirm whether such adjunctive MMP inhibition can translate into improved clinical outcomes.

## 7. Conclusions and Future Directions

In summary, the present review underscores the pivotal role of HDTs in the mitigation of the substantial burden caused by infectious diseases, with particular emphasis on mycobacterial infection and Mtb/HIV coinfection. Moreover, it delves into innovative HDTs based on protease inhibitors and their clinical implications as adjunctive therapeutic regimens for the management of TB and HIV coinfection. These novel strategies include the repurposing of SQV (an HIV protease inhibitor) and the manipulation of Csts (the natural cathepsin inhibitors). The present review emphasizes that HDTs represent an evolving approach to managing bacterial infection, mainly by restoring the compromised host immune responses. The transition from Mtb infection to active disease is influenced by coinfection and immunosuppression, as observed in HIV coinfection. Furthermore, HDTs have been shown to boost immune responses and alleviate immunopathology by targeting host immune and inflammatory pathways, resulting in beneficial effects on outcomes of bacterial and viral treatments. A variety of HDTs have been developed to combat pathogens by targeting multiple cellular and intracellular mechanisms required for microbial replication and pathogenesis. Consequently, HDTs may also reduce pathogen transmission within the community and improve clinical outcomes. This suggests that HDTs could be a game-changing strategy in treating infectious diseases.

## Figures and Tables

**Figure 1 microorganisms-13-01040-f001:**
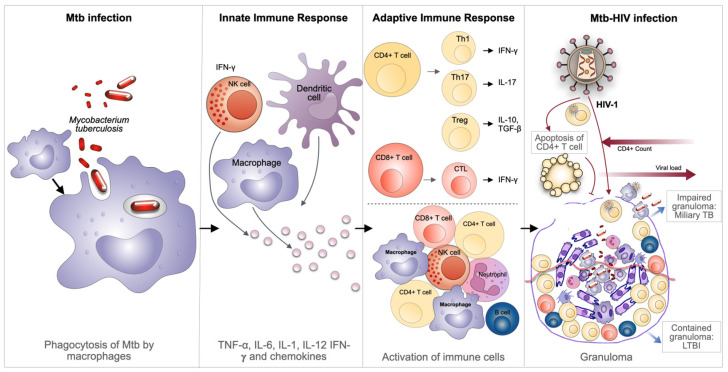
Immune response in TB and during HIV coinfection. Alveolar macrophages, as professional phagocytes, constitute the first innate immune cells to restrict the infection. However, Mtb can establish intracellular survival niches within endosomes. The innate immune response is executed by coordinating the activity of macrophages, NK cells, and DC, which secrete an array of effector cytokines (e.g., TNF-α, IFN-γ, IL-1) and various chemokines. This process recruits CD4^+^ and CD8^+^ T-cells to release their specific cytokines, such as IFN-γ, which activate infected macrophages to a more bactericidal state. This, in turn, triggers a concerted immune cascade that ultimately results in granuloma formation. Apoptosis of cells infected with Mtb is promoted by TNF-α following its activation by IFN-γ or by the cytotoxic activity of CTLs. This process contributes to the control of the infection in a latent form within the host. However, the coinfection with HIV results in a systemic immunodepression, which is characterized by a dramatic loss of functional CD4^+^ T-cells by apoptosis. This depletion of CD4^+^ T-cells leads to a deficient supply of these lymphocytes to the granuloma, contributing to the disruption of that solid architecture. Figure created using Keynote for Mac.

**Figure 2 microorganisms-13-01040-f002:**
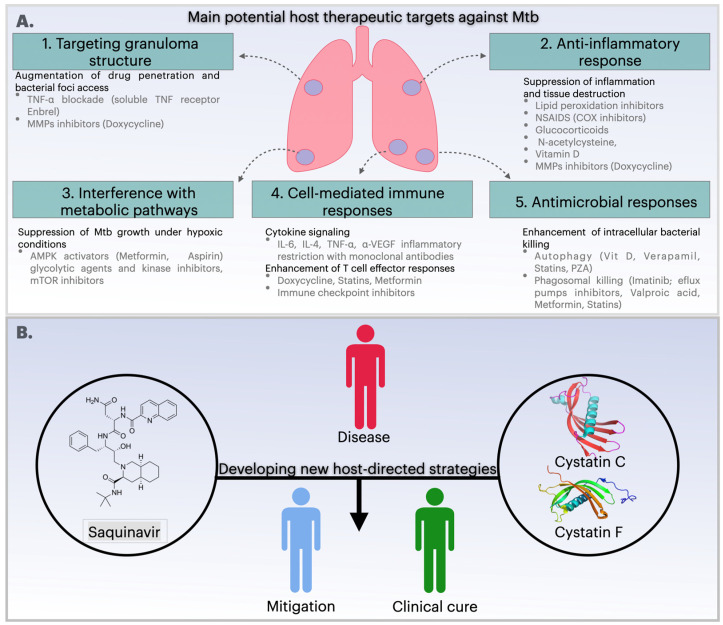
Host-directed therapies (HDT) against *Mycobacterium tuberculosis*. (**A**) Main HDT used to improve the outcome of TB. (**1**) HDT can affect granuloma integrity to improve antibiotic penetration and access to bacterial foci. (**2**) HDT can act by suppressing the proinflammatory responses that contribute to tissue damage during TB. (**3**) The interference with metabolic pathways is a promising strategy for suppression Mtb growth under hypoxic conditions. (**4**) HDT can target cell-mediated immune responses by enhancing T-cell responses through doxycycline, statins and metformin. (**5**) Autophagy inducing drugs such as vitamin D, verapamil, and statins contribute the intracellular killing of Mtb in lysosomes. (**B**) Development of new host-directed strategies targeting Mtb. The investigation of SQV as a repurposed drug for the control of Mtb infection and the targeting of cystatins such as cystatin C (CstC) and cystatin F (CstF) in human macrophages may represent a new potential adjuvant therapy for TB. AMPK, 5′ adenosine monophosphate-activated protein kinase; NSAID, non-steroidal anti-inflammatory drug; COX, cyclooxygenase; mTOR, mammalian target of rapamycin; PZA, pyrazinoic acid; MMPs, Metalloproteinase inhibitors; VEGF, vascular endothelial growth factor. Figure created using Keynote for Mac.

**Figure 3 microorganisms-13-01040-f003:**
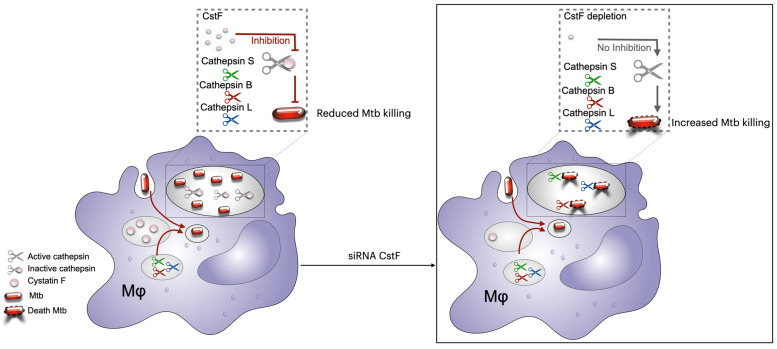
CstF depletion in Mtb-infected macrophages improves the proteolytic activity of cathepsins L, B, and S, thereby significantly impacting the intracellular killing of the pathogen. Interference in CstF expression could restore the basal levels of proteolytic activity detected prior to infection with Mtb. Figure created using Keynote for Mac.

**Figure 4 microorganisms-13-01040-f004:**
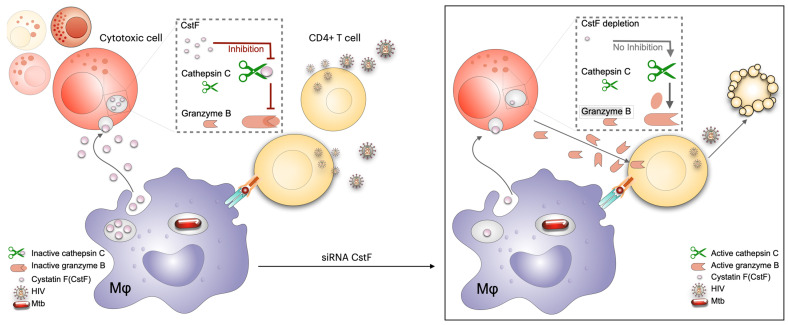
Impact of CstF depletion from Mtb-infected macrophages on the cytotoxic activity of lymphocytes. During the coinfection of Mtb-infected macrophages with lymphocytes infected with HIV, CstF depletion enhances cathepsin C proteolytic activity, which in turn activates granzyme B. This process results in a notable decrease in the viral load. Figure created using Keynote for Mac, adapted from [166].

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
