# Peer review of "Host-Directed Therapies Based on Protease Inhibitors to Control *Mycobacterium tuberculosis* and HIV Coinfection"

_microorganisms, 2025, doi:10.3390/microorganisms13051040_

Round 1

Reviewer 1 Report

Comments and Suggestions for Authors

Reviewer’s Comments

 General assessment:

This review article addresses an important and timely topic, focusing on the role of host-directed therapies (HDTs), specifically protease inhibitors, in managing Mycobacterium tuberculosis (Mtb) and HIV coinfection. The authors have presented a comprehensive overview of the dual burden imposed by TB and HIV, with a focus on the therapeutic promise of agents like saquinavir and cystatins in modulating host immune responses. The article is informative, well-structured in terms of topic organization, and contributes valuable insights into adjunctive treatment strategies that may improve clinical outcomes in coinfected individuals.

However, the manuscript would benefit significantly from revisions in scientific writing quality and addressing the following comments.

Specific comments:

  1. Scientific context and justification:

The manuscript effectively outlines the global health challenges posed by TB and HIV coinfection, particularly in the face of rising antimicrobial resistance and limited treatment options. The emphasis on the limitations of current treatment regimens and the rationale for host-directed therapies is commendable.

  1. Discussion of protease inhibitors:

The authors provide a detailed discussion of saquinavir (SQV) and cystatins (CstC and CstF) as HDT candidates. The mechanistic insights into SQV’s role in modulating macrophage function and enhancing TB control are well-articulated. Similarly, the role of cystatins in improving host immune responses against Mtb is clearly described. The manuscript successfully integrates the evidence reported by various studies supporting these mechanisms. However, the discussion could be enhanced by providing more detailed comparative insights into the efficacy of these agents versus conventional anti-TB and antiretroviral therapies, as well as highlighting potential safety concerns or limitations in their clinical translation.

  1. Clarity on the mechanistic role of HDTs:

The review convincingly outlines how HDTs can modulate host immunity, reduce immunopathology, and enhance treatment outcomes in coinfected individuals. The manuscript rightly emphasizes the repurposing of existing protease inhibitors and the novel therapeutic potential of targeting host pathways. Nonetheless, a deeper discussion on how these therapies interact with standard drug regimens, particularly in terms of pharmacokinetics, drug-drug interactions, and host safety, would add value.

  1. Scope and breadth:

While the review focuses on protease inhibitors, the overall scope could be broadened by briefly discussing other promising HDTs (e.g., autophagy modulators, immune checkpoint inhibitors, metabolic reprogramming agents, etc.) in a comparative context. This would enhance the article’s comprehensiveness and attract a wider readership.

  1. Language and scientific writing:

The manuscript requires substantial revision to improve language quality, including grammar, sentence structure, and scientific tone. Several sections contain informal phrasing and grammatical inconsistencies that detract from the scientific rigor expected in a high-quality review.

Recommendation: A thorough professional English editing is strongly recommended to ensure that the manuscript meets the standards of scientific literature.

Additional suggestions:

A graphical abstract can be included to illustrate the mechanisms by which protease inhibitors modulate host responses during TB/HIV coinfection.

Discuss the translational challenges of HDTs, including regulatory hurdles, biomarker identification for patient stratification, and cost-effectiveness in low-resource settings.

Provide a table summarizing ongoing or completed clinical trials involving protease inhibitors for TB or HIV to add translational relevance.

Overall Recommendation:

The manuscript presents an important and underexplored area of research with high potential impact. However, before acceptance, the authors should:

  1. Revise the manuscript for language and grammatical accuracy.
  2. Include comparative analysis and address translational challenges.

Upon addressing these revisions, the manuscript will offer a valuable and timely contribution to the field of infectious disease therapeutics.

Author Response

General assessment:

This review article addresses an important and timely topic, focusing on the role of host-directed therapies (HDTs), specifically protease inhibitors, in managing Mycobacterium tuberculosis (Mtb) and HIV coinfection. The authors have presented a comprehensive overview of the dual burden imposed by TB and HIV, with a focus on the therapeutic promise of agents like saquinavir and cystatins in modulating host immune responses. The article is informative, well-structured in terms of topic organization, and contributes valuable insights into adjunctive treatment strategies that may improve clinical outcomes in coinfected individuals.

However, the manuscript would benefit significantly from revisions in scientific writing quality and addressing the following comments.

Specific comments:

  1. Scientific context and justification:

The manuscript effectively outlines the global health challenges posed by TB and HIV coinfection, particularly in the face of rising antimicrobial resistance and limited treatment options. The emphasis on the limitations of current treatment regimens and the rationale for host-directed therapies is commendable.

R: Regarding the reviewer's comment, we have extended the section concerning the current treatment regimens, drug interactions, and included the latest developments with the introduction of the BPaL regimen for drug-resistant TB. We also expand on the adverse effects of treating drug-resistant TB, the financial impact on the affected households, the ethical concerns of preventive treatment of latently infected individuals, and the risk of promoting the evolution of XDR-TB strains. All these changes are marked in yellow in section 3. We believe this section now better illustrates why this curable disease still faces so many challenges that warrant the investment in innovative strategies such as HDTs.

  1. Discussion of protease inhibitors:

The authors provide a detailed discussion of saquinavir (SQV) and cystatins (CstC and CstF) as HDT candidates. The mechanistic insights into SQV’s role in modulating macrophage function and enhancing TB control are well-articulated. Similarly, the role of cystatins in improving host immune responses against Mtb is clearly described. The manuscript successfully integrates the evidence reported by various studies supporting these mechanisms. However, the discussion could be enhanced by providing more detailed comparative insights into the efficacy of these agents versus conventional anti-TB and antiretroviral therapies, as well as highlighting potential safety concerns or limitations in their clinical translation.

R: We have expanded the sections concerning HDTs and protease inhibitors. We mention examples of protease inhibitors being tested directly against the bacteria to illustrate some of the related challenges, such as detrimental interactions with human proteases (in yellow, section 5). We also mention the problems related to low serum availability of SQV, negative interactions with the standard TB therapy. We believe it is now clearer that a solution such as SQV would have to come with a drug delivery strategy that can circumvent those problems (Section 6.1, in yellow). For cystatins, all the evidences come from in vitro assays, so for now, clinical applicability can only be speculated.

  1. Clarity on the mechanistic role of HDTs:

The review convincingly outlines how HDTs can modulate host immunity, reduce immunopathology, and enhance treatment outcomes in coinfected individuals. The manuscript rightly emphasizes the repurposing of existing protease inhibitors and the novel therapeutic potential of targeting host pathways. Nonetheless, a deeper discussion on how these therapies interact with standard drug regimens, particularly in terms of pharmacokinetics, drug-drug interactions, and host safety, would add value.

R: As answered in the previous comments, we have followed the reviewer's suggestion and expanded on the topics of drug interactions, both in the standard regimens and in the newly proposed therapies. We mention how Mtb-directed protease inhibitors can negatively impact the host, the challenges posed by SQV low serum availability and interaction with rifampicin, and also in a new section about matrix metalloproteinases, we also address conflicting reports on the benefits and challenges of targeting these molecules.

  1. Scope and breadth:

While the review focuses on protease inhibitors, the overall scope could be broadened by briefly discussing other promising HDTs (e.g., autophagy modulators, immune checkpoint inhibitors, metabolic reprogramming agents, etc.) in a comparative context. This would enhance the article’s comprehensiveness and attract a wider readership.

R: We agree with the reviewer that a broader analysis of HDTs would be interesting. This, however, has already been published recently in a review more focused on anti-inflammatory HDTs (10.3389/fmed.2022.970408), and our objective was to focus on protease modulation as a strategy for HDTs and to increase awareness of the potential of these molecules. To provide context, we have already mentioned other types of HDTs and immunomodulatory strategies, and we also mention protease modulation in the context of pathogen-targeted strategies (and we expanded that in this revised version). To broaden the interest in this review, we have now also included a section on matrix metalloproteinases, since there are several MMPs being explored as targets, and some drugs and clinical trials with very recent results.

  1. Language and scientific writing:

The manuscript requires substantial revision to improve language quality, including grammar, sentence structure, and scientific tone. Several sections contain informal phrasing and grammatical inconsistencies that detract from the scientific rigor expected in a high-quality review.

Recommendation: A thorough professional English editing is strongly recommended to ensure that the manuscript meets the standards of scientific literature.

R: The English editing was performed as recommended to improve the language quality, to remove more colloquial terms, and to use more scientifically appropriate terms. We hope the reviewer finds this version to be clearer and more rigorous.

Additional suggestions:

A graphical abstract can be included to illustrate the mechanisms by which protease inhibitors modulate host responses during TB/HIV coinfection.

Discuss the translational challenges of HDTs, including regulatory hurdles, biomarker identification for patient stratification, and cost-effectiveness in low-resource settings.

Provide a table summarizing ongoing or completed clinical trials involving protease inhibitors for TB or HIV to add translational relevance.

Overall Recommendation:

The manuscript presents an important and underexplored area of research with high potential impact. However, before acceptance, the authors should:

  1. Revise the manuscript for language and grammatical accuracy.
  2. Include comparative analysis and address translational challenges.

Upon addressing these revisions, the manuscript will offer a valuable and timely contribution to the field of infectious disease therapeutics.

R: We thank the reviewer for all the suggestions. We have substantially improved the manuscript with language corrections and with new sections of text as previously mentioned. We also introduce a graphical abstract as suggested. We now mention more results from clinical trials, but since most HDT strategies mentioned are still in pre-clinical stages, the evidence is still scarce. We believe this version is significantly improved and better addresses most of the reviewer comments.

Reviewer 2 Report

Comments and Suggestions for Authors

The manuscript addresses an important and timely issue in infectious disease research. Tuberculosis remains a major global health challenge, particularly in the context of HIV coinfection, which exacerbates disease progression and complicates treatment. The main points of the review are supplemented with appropriate schematic illustrations. The manuscript presents a well-structured review based on existing literature and experimental findings. The references are generally appropriate, covering key studies in the field. However, I have some minor comments.

Minor comments:

Line 73. “PAMPS” – “PAMPs”.

Lines 130-131. “The coinfection of Mtb with HIV therefore appears to be the most significant risk factor for the progression to active TB.” It would be more accurate to state "one of the most significant risk factors".

Lines 282-283. “Administration of type I interferon (IFN) has also been used for the treatment of chronic Hepatitis C (HCV) infection or COVID-19 [61–63].” – I believe reference [62] discusses type III interferon. Please verify and either replace the reference or rephrase the sentence accordingly.

Lines 285-286. “virus infection (RSV) infection” – Repetition of “infection”.

Line 326. “at 1.25 mg supplementation” ─ It is unclear why 1.25 mg is mentioned. The cited reference [79] states 2.5 mg. Please verify and correct if necessary.

Line 390. “In (1995)” − “In 1995”.

Line 480. “cytoxicity” ─ “cytotoxicity”.

Author Response

The manuscript addresses an important and timely issue in infectious disease research. Tuberculosis remains a major global health challenge, particularly in the context of HIV coinfection, which exacerbates disease progression and complicates treatment. The main points of the review are supplemented with appropriate schematic illustrations. The manuscript presents a well-structured review based on existing literature and experimental findings. The references are generally appropriate, covering key studies in the field. However, I have some minor comments.

Minor comments:

Line 73. “PAMPS” – “PAMPs”.

Lines 130-131. “The coinfection of Mtb with HIV therefore appears to be the most significant risk factor for the progression to active TB.” It would be more accurate to state "one of the most significant risk factors".

Lines 282-283. “Administration of type I interferon (IFN) has also been used for the treatment of chronic Hepatitis C (HCV) infection or COVID-19 [61–63].” – I believe reference [62] discusses type III interferon. Please verify and either replace the reference or rephrase the sentence accordingly.

Lines 285-286. “virus infection (RSV) infection” – Repetition of “infection”.

Line 326. “at 1.25 mg supplementation” ─ It is unclear why 1.25 mg is mentioned. The cited reference [79] states 2.5 mg. Please verify and correct if necessary.

Line 390. “In (1995)” − “In 1995”.

Line 480. “cytoxicity” ─ “cytotoxicity”.

R: We thank the reviewer for the comments and all the corrections. We have followed all corrections with the text marked in yellow in the manuscript.

Reviewer 3 Report

Comments and Suggestions for Authors

The article is extensive, but it fails to tie mechanisms. It reads more like a patchwork of detailed independent pathways, that do not cohesively appear particular to M.tuberculosis infection. When authors try to incorporate Mtb infection, it feels disconnected, vague, and conjectural.

Overall, it lacks cohesiveness and fails to provide concrete connections and coherence.

Author Response

The article is extensive, but it fails to tie mechanisms. It reads more like a patchwork of detailed independent pathways, that do not cohesively appear particular to M.tuberculosis infection. When authors try to incorporate Mtb infection, it feels disconnected, vague, and conjectural.

Overall, it lacks cohesiveness and fails to provide concrete connections and coherence.

R: We have substantially improved the manuscript and introduced new sections related to challenges to the current TB therapy, Mtb proteases that are being explored as targets, and other host proteases that are being addressed as HDTs in tuberculosis.

Reviewer 4 Report

Comments and Suggestions for Authors

This review by Mandal et al. investigates the potential use of protease inhibitors in host-directed therapies (HDT) to treat Mycobacterium tuberculosis (MTB) and HIV co-infection. Given the ongoing global challenge posed by the co-epidemics of MTB and HIV, the subject remains timely and of critical interest. The paper provides an excellent overview of current methods and highlights the significance of employing protease inhibitors in treatment. However, a lack of originality and inadequate distinction from similar reviews (e.g., PMID: 34421929) preclude its acceptance in its current form. Below are my comments, corrections, and suggestions:

Major issues/comments: 

  1. In my opinion, the current title does not accurately reflect the discussion on both viral and host protease inhibitors used in the therapeutic strategies. I recommend changing the title to "Host and Viral Directed Therapies Based on Protease Inhibitors to Control Mycobacterium Tuberculosis and HIV Coinfection" to represent the dual aspects of the review better. The authors should also discuss the potential for developing dual inhibitors, i.e., compounds that inhibit HIV (or MTB) protease and the host’s proteases.
  2. The manuscript should differentiate between the roles of viral protease inhibitors, such as Saquinavir, primarily targeting HIV viral replication, and host protease effectors, such as Cystatins C and F (that target Cathepsin), aligning with host-directed therapies for MTB. Even though the rationale for Cystatins is well within host-directed therapies, the situation with Saquinavir and other similar compounds is more directly involved with viral inhibition in HIV and not MTB. For further clarification, the authors may define the respective functions and operations of HIV and MTB proteases and compare their structural and functional characteristics.
  3. The discussion of the role of proteolytic enzymes in maintaining host cellular homeostasis is too general. The authors should expand this section by:
    • Providing the classes of host proteases (e.g., serine proteases vs. metalloproteinases). The authors should discuss specific examples of host proteases implicated in both diseases. To the best of my knowledge, in addition to Cathepsins, other host proteases such as MMP-1, MMP-2, and MMP-9 have been implicated in the development of tuberculosis (please see the following example papers: PMID: 21519144 and PMID: 31199047).
    • Describing their mechanisms of action within host cells, particularly their roles in immune regulation and/or cellular homeostasis.
    • Explaining the specific roles these proteases play in the pathogenesis and immune responses against Mycobacterium tuberculosis and HIV.
    • Discussing how manipulating these proteases could lead to novel therapeutic strategies for preventing or treating infections.
  4. The review would benefit from a deeper analysis of the clinical relevance and the current stage of research for each class of protease inhibitors. A comparison between the efficacies of viral and host protease inhibitors in clinical and pre-clinical settings would be beneficial.
  5. There is an emphasis on the potential benefits without adequate discussion of the limitations, side effects, or resistance issues associated with using protease inhibitors, especially in complex co-infections like TB-HIV.
  6. While the manuscript covers a broad topic, this wide scope results in a lack of detail in certain areas, such as the specific biochemical pathways involved and direct comparisons between host and viral protease inhibitors.

Minor Comments:

  • LL512-513: There is a discrepancy in fold type and size.
  • It is unclear whether the figures have been obtained from previously published papers or created for the needs of this manuscript. If the authors have adapted or reproduced figures from other sources, then they should include relevant references. For figures created specifically for this manuscript, the software used should be specified.

Author Response

This review by Mandal et al. investigates the potential use of protease inhibitors in host-directed therapies (HDT) to treat Mycobacterium tuberculosis (MTB) and HIV co-infection. Given the ongoing global challenge posed by the co-epidemics of MTB and HIV, the subject remains timely and of critical interest. The paper provides an excellent overview of current methods and highlights the significance of employing protease inhibitors in treatment. However, a lack of originality and inadequate distinction from similar reviews (e.g., PMID: 34421929) preclude its acceptance in its current form. Below are my comments, corrections, and suggestions:

Major issues/comments: 

  1. In my opinion, the current title does not accurately reflect the discussion on both viral and host protease inhibitors used in the therapeutic strategies. I recommend changing the title to "Host and Viral Directed Therapies Based on Protease Inhibitors to Control Mycobacterium Tuberculosis and HIV Coinfection" to represent the dual aspects of the review better. The authors should also discuss the potential for developing dual inhibitors, i.e., compounds that inhibit HIV (or MTB) protease and the host’s proteases.

R: With this review, we focused on proposed solutions targeting host proteases and their potential for an HDT. We mention viral protease inhibitors such as SQV because they were shown to modulate human proteases. As such, despite addressing inhibitors of viral proteases, our interest is not in how they directly interfere with the virus, which has been substantially explored already, but the surprising fact that some of these inhibitors actually activate human proteases. Nonetheless, we have extended the section addressing potential pathogen-directed strategies to provide more context.

  1. The manuscript should differentiate between the roles of viral protease inhibitors, such as Saquinavir, primarily targeting HIV viral replication, and host protease effectors, such as Cystatins C and F (that target Cathepsin), aligning with host-directed therapies for MTB. Even though the rationale for Cystatins is well within host-directed therapies, the situation with Saquinavir and other similar compounds is more directly involved with viral inhibition in HIV and not MTB. For further clarification, the authors may define the respective functions and operations of HIV and MTB proteases and compare their structural and functional characteristics.

R: As mentioned in our response to the previous comment, we address SQV not as an inhibitor of the viral protease and of viral replication, but as a molecule that was reported to be repurposable to target human proteases (human cathepsins). As such, we have only included a brief section providing context on protease inhibitors targeting the pathogen, so the reader understands the origin of these molecules. However, it is not our intention to focus on their pathogen-directed activity. As such, SQV and the cystatin modulators all fall into the same category of host protease effectors. Still, we have expanded the section addressing inhibition of microbial proteases, especially concerning Mtb. We have further provided more evidence concerning the clinical use of some of these solutions, adverse effects, and interactions with current therapy.

  1. The discussion of the role of proteolytic enzymes in maintaining host cellular homeostasis is too general. The authors should expand this section by:
    • Providing the classes of host proteases (e.g., serine proteases vs. metalloproteinases). The authors should discuss specific examples of host proteases implicated in both diseases. To the best of my knowledge, in addition to Cathepsins, other host proteases such as MMP-1, MMP-2, and MMP-9 have been implicated in the development of tuberculosis (please see the following example papers: PMID: 21519144 and PMID: 31199047).
    • Describing their mechanisms of action within host cells, particularly their roles in immune regulation and/or cellular homeostasis.
    • Explaining the specific roles these proteases play in the pathogenesis and immune responses against Mycobacterium tuberculosis and HIV.
    • Discussing how manipulating these proteases could lead to novel therapeutic strategies for preventing or treating infections.

R: We thank the reviewer for these suggestions. We have improved the manuscript accordingly, mentioning the different classes of proteases in section 5 and their preponderance in humans, and we have introduced an entire section about MMPs, as these are of great relevance to this subject (section 6.3, marked in yellow).

  1. The review would benefit from a deeper analysis of the clinical relevance and the current stage of research for each class of protease inhibitors. A comparison between the efficacies of viral and host protease inhibitors in clinical and pre-clinical settings would be beneficial.

R: We agree with the reviewer that more information on translatable aspects of HDT and evidence from clinical application is of great interest. We have included information about this in every section. Concerning inhibitors of Mtb-proteases, we mention the challenges of negative interactions with human proteases. With protease inhibitors for HDTs, we now mention how SQV application faces several challenges, such as low serum availability, interaction with TB drug regimen, and the necessity for incorporating this solution in an advanced drug-delivery method. For MMPs, we discuss the conflicting evidence from different proposed solutions and concluded clinical trials. For cystatin-targeting in TB, the evidence is still pre-clinical.

  1. There is an emphasis on the potential benefits without adequate discussion of the limitations, side effects, or resistance issues associated with using protease inhibitors, especially in complex co-infections like TB-HIV.

R: As already mentioned in the previous point, we followed the reviewer’s suggestion and now included information on the several challenges faced by the current TB regimen, on the pitfalls of using pathogen-targeted protease inhibitors, and on the difficulties already reported from clinical and pre-clinical studies on the use of SQV and MMP inhibitors. These changes are marked in yellow.

  1. While the manuscript covers a broad topic, this wide scope results in a lack of detail in certain areas, such as the specific biochemical pathways involved and direct comparisons between host and viral protease inhibitors.

R: As we previously mentioned, our discussion of viral protease inhibitors is only present to provide context to the origins of these inhibitors, which are now proposed for the modulation of host proteases. In many of these cases, there is little mechanistic evidence for how this occurs, and thus, we have decided to use this review to compile and provide awareness to potential candidates that should be further explored to understand and prove their applicability as HDTs in TB and in the context of coinfection with HIV. For more detailed analysis on the role of human proteases and endogenous inhibitors, there are other reviews, as the reviewer precisely mentioned, such as PMID: 34421929.

Minor Comments:

  • LL512-513: There is a discrepancy in fold type and size.
  • It is unclear whether the figures have been obtained from previously published papers or created for the needs of this manuscript. If the authors have adapted or reproduced figures from other sources, then they should include relevant references. For figures created specifically for this manuscript, the software used should be specified.

R: We thank the reviewer for these corrections. We corrected the discrepancy in fold type and size. We made extensive improvements to the language, and we believe the manuscript is now clearer and more consistent. We now mention the software used to create the figures for this manuscript. All figures were created by the authors for this manuscript using Keynote for Mac. Adaptations are mentioned in the legend with the respective reference.

Round 2

Reviewer 3 Report

Comments and Suggestions for Authors

This review is still all over the place. It does not focus on the proposed (in my opinion misleading) title.

Reviewer 4 Report

Comments and Suggestions for Authors

The authors have addressed the vast majority of my concerns, and the manuscript has been significantly improved after the first round of revisions. I believe the manuscript can be accepted in its current form.